# Selective Laser Sintering (SLS) and Post-Processing of Prosopis Chilensis/Polyethersulfone Composite (PCPC)

**DOI:** 10.3390/ma13133034

**Published:** 2020-07-07

**Authors:** Aboubaker I. B. Idriss, Jian Li, Yangwei Wang, Yanling Guo, Elkhawad A. Elfaki, Shareef A. Adam

**Affiliations:** 1College of Mechanical and Electrical Engineering, Northeast Forestry University, Harbin 150040, China; aboubakerbolad@yahoo.com (A.I.B.I.); wywkly@126.com (Y.W.); guo.yl@hotmail.com (Y.G.); 2Department of Mechanical Engineering, Faculty of Engineering Science, University of Nyala, P.O. Box 155, Nyala 11111, Sudan; shadiahcom@yahoo.com; 3Mechanical Engineering Department, College of Engineering, The University of Bisha, Bisha 61922, Saudi Arabia; khawad.2002@gmail.com; 4Department of Mechanical Engineering, College of Engineering, Sudan University of Science and Technology, Khartoum 11113, Sudan

**Keywords:** selective laser sintering process, Prosopis chilensis powder (PCP), polyethersulfone (PES) powder, mechanical properties, post-processing, dimensional precision analysis

## Abstract

The range of selective laser sintering (SLS) materials is currently limited, and the available materials are often of high cost. Moreover, the mechanical strength of wood–plastic SLS parts is low, which restricts the application of a SLS technology. A new composite material has been proposed to address these issues, while simultaneously valorizing agricultural and forestry waste. This composite presents several advantages, including reduced pollution associated with waste disposal and reduced CO_2_ emission with the SLS process in addition to good mechanical strength. In this article, a novel and low-cost Prosopis chilensis/polyethersulfone composite (PCPC) was used as a primary material for SLS. The formability of PCPC with various raw material ratios was investigated via single-layer experiments, while the mechanical properties and dimensional accuracy of the parts produced using the various PCPC ratios were evaluated. Further, the microstructure and particle distribution in the PCPC pieces were examined using scanning electron microscopy. The result showed that the SLS part produced via 10/90 (wt/wt) PCPC exhibited the best mechanical strength and forming quality compared to other ratios and pure polyethersulfone (PES), where bending and tensile strengths of 10.78 and 4.94 MPa were measured. To improve the mechanical strength, post-processing infiltration was used and the PCPC-waxed parts were enhanced to 12.38 MPa and 5.73 MPa for bending and tensile strength.

## 1. Introduction

Selective laser sintering (SLS) is a three-dimensional (3D) additive manufacturing (AM) technology that was first proposed by Deckard in 1988 [1,2,3,4,5]. The technique involves the use of a laser beam to directly fuse a powdered raw material over a large area, where successive layers are deposited based on a three-dimensional stereolithography (STL)-formatted computer-aided design (CAD) model developed without expensive equipment [6,7]. A wide variety of powdered raw materials can be fused and melted during laser sintering [6,7,8,9]. SLS technology has many advantages over other 3D technologies and can be used to manufacture complex parts without additional support [10,11,12]. The unsintered materials are reusable and recyclable, and the manufactured parts are often more accurate than those produced using other techniques. SLS can be applied widely for the manufacture of construction, medical, automobile, and industrial products, where composite materials and nanocomposite can be used [13,14,15,16,17,18,19,20]. SLS technology is rapidly developing and has been used in a variety of composite product applications, including wood-based materials and investment casting. To further improve the SLS technology, there is a growing need to research and develop novel materials, as well as to investigate the effects of particle size and mixture ratios on the mechanical properties of sintered parts [21,22,23]. The quality of SLS-manufactured parts is often optimized using specialized composite materials. A wide variety of materials can be used with SLS technology, including ceramics, polycarbonates, polymers, metals, and composites, but the high price of these materials has restricted a further application of SLS [24]. Consequently, a reduction in the cost of these raw materials is expected to promote wider implementation of the technology [13].

Numerous studies have investigated the processing of green biomass composites via SLS. Guo et al. at the Northeast Forestry University (NEFU) proposed the novel use of low-cost natural agricultural and forestry waste as a primary SLS material to address the limited availability and the high cost of materials. Furthermore, this approach had a lower environmental impact, specifically carbon dioxide emission, and widened the range of available SLS materials. Other studies have evaluated the use of sintered sisal fiber/polyethersulfone (PES) composite (SFPC), rice husk powder/co-polyethersulfone (PES) composite (RHPC), walnut shell/co-PES composite, bamboo–plastic composite, and wood–plastic composites. The range of SLS materials is limited, and current composites are associated with quality defects. Therefore, a suitable composite powder mixture ratio, diameter range, and optimal processing parameters must be established to ensure the production of high quality sintered parts.

Prosopis chilensis wood powder (PCP) is a green biomass material produced from renewable forestry waste. PCP is a sustainable renewable waste source with a lower cost and lower carbon emission than bamboo, other wood powders, and rice husk. Further, the Prosopis chilensis tree can be cultivated in highly saline desert soils, as it is naturally occurring in the Arabian Peninsula, the Jordan Valley, Egypt (including Sinai), Sudan, some Maghreb countries (Libya, Algeria, and Morocco), the Horn of Africa, Iran, and India. Further, the Prosopis chilensis tree is a significant issue in most agricultural areas in Sudan, as the species occupies large surface areas and can negatively affect the fertility of the soil where areas become unfit for agriculture. The Sudanese government has attempted to mass burn Prosopis chilensis, thus causing environmental damage. Hence, the experimental objective of this study is to valorize the biomass of the Prosopis chilensis tree in AM technology via SLS instead of burning it for disposal. The Prosopis chilensis composite wood is expected to have a positive economic and environmental impact, where PCP was added to PES powder to produce a novel Prosopis chilensis/PES composite (PCPC) SLS material. The composite offered numerous benefits, including high hardness and good mechanical strength, where the sintered parts exhibited superior smooth surface roughness, anti-aging properties, and dimensional stability compared to other wood composite powders. Moreover, the unsintered powder was recyclable and low in cost. Furthermore, one of the expected potential environmental benefits of this product is reducing the influence of pressure on the forest sector in Sudan that is cut for the use of home furniture purposes. Prosopis chilensis composite can be used for parquet floors, doors, furniture, construction, and other purposes.

This novel study presents the first use of PCP in the SLS process, where PCP and PES powders were used as the feedstock for the production of low-cost environmentally friendly sintered parts with good mechanical strength. PCPC composites with various mixture ratios were prepared, where PCP/PES ratios of 10/90, 15/85, and 20/80 (wt/wt) were evaluated. The density, mechanical properties, dimensional accuracy, and surface roughness of the sintered parts were analyzed, where the mechanical properties were compared to previously reported results. The post-processing methods were used to further improve the mechanical properties of the PCPC SLS parts. Post-processing infiltration with wax was introduced, where the SLS parts were immersed with wax. In addition, the bending strength, tensile strength, and surface roughness were investigated before and after post-processing. The main materials used in the post-processing treatment method are paraffin industrial wax and pool wax.

## 2. Materials and Method

### 2.1. Materials

#### 2.1.1. PCP

Prosopis chilensis timber samples were collected from the White Nile State of Sudan. The species was authenticated by the National Natural Science Foundation of China (Grant No. 51905084). The Prosopis chilensis samples were dried at room temperature, processed into PCP using crushing machinery (Jiangsu Guibao Group Co., Ltd., Beijing, China) and sieved using standard vibrating sieves (Figure 1). The PCP was sifted through a 100 mesh screen via an intensive shaking procedure using a vibrating sifter (Model ZS 350 Jiangsu Guibao Group Co., Ltd., Beijing, China). To remove powder agglomerates as well as obtain a PCP particle size range of less than 0.125 mm. The PCP was dried and cooled to room temperature for PCPC synthesis. The morphologies of the PCP and PES powder parts are shown in Figure 5.

#### 2.1.2. Prosopis Chilensis/PES Composite Powder (PCPC)

The primary materials included PCP and PES powder. PES is a thermoplastic polymer and was obtained from Shanghai TianNian Material Technology (Shanghai, China). PES is a type of thermoplastic polymer with excellent comprehensive properties and a stable performance temperature range (−100 °C to 200 °C), which was developed by the British Imperial Chemical Industry Group in the 1970s (Knoell et al., 1999). Moreover, PES is one of the most widely used materials for special engineering plastics. The molecular structure of PES includes three groups, namely, a sulfone, ether, and phenylene group. The PCP was manually mixed with the PES powder and mechanically mixed using an SHR-10A high-speed mixer to obtain uniform distribution of the powders. PCPC with PCP/PES ratios of 10/90, 15/85, and 20/80 (wt/wt) were prepared and used for SLS. In this study, the glass transition temperature (Tg) of PES and PCP were found through the DSC test; based on the experiments, the results determined the optimum sintering windows, specifically the processing temperature to produce the PCPC pieces parts. The DSC curves of the PES and PCP powders are shown in Figure 6, where Tg of PES and PCP should be not more than 88 and 89 °C. The PCPC preparation flow chart is illustrated in Figure 1 and Figure 2.

### 2.2. Methodology

#### 2.2.1. Selective Laser Sintering

The various sintered PCPC parts were produced using an AFS-360 SLS machine rapid prototyping machine made by the (Beijing Longyuan Technology, Ltd. Beijing, China), where the SLS process is illustrated in Figure 3. The main processing parameters included a layer thickness of 0.1 mm, scanning speed of 2000 mm/s, scan spacing of 0.2 mm, preheating temperature of 82 °C, processing temperature of 75 °C, and laser power of 14 W. Furthermore, the laser wavelength is 10.6 μm.

#### 2.2.2. Post-Processing

To further improve the mechanical properties and surface quality of SLS sintered parts, the post-processing infiltration with wax was introduced, and the post-processing flow chart is shown in Figure 4.

The PCPC parts were removed from the AFS-360 machine, cleaned carefully, and placed into an electrical heating thermostat for insulation. Following the insulation, the PCPC parts were slowly immersed in a wax pool to allow for wax infiltration via capillarity action within the voids between the particles. The melted wax led to excellent bonding of the PCPC particles. The mechanical strength of the PCPC SLS parts expected to improve by 13% and 14% for bending and tensile strength with post-processing due to a decrease in the voids fraction between the PCP and PES particles and an increase in the density of SLS part. The infiltrated parts were slightly larger (0.1 mm) in all three dimensions (X, Y, and Z directions). This increase was small, confirming that post-processing had little effect on the accuracy of the SLS parts. The surface of the post-processed parts was smoother where the surface roughness value decreased. The post-processing process is illustrated in Figure 4.

#### 2.2.3. Scanning Electron Microscopy (SEM)

PCP and the PES powder were scanned using a FEI Quanta 200 SEM scanning electron microscope (Hewlett-Packard Company, Amsterdam, Netherlands). The morphologies of the PCP and PES powder parts are shown in Figure 5.

#### 2.2.4. Differential Scanning Calorimetry

The glass transition temperatures of the PCP and PES powder (5 mg) were tested using a differential scanning calorimeter (DSC) American Pyris Diamond differential scanning calorimeter (Perkin Elmer, Waltham, MA, USA) in a testing temperature range of 20 °C to 240 °C at a heating rate of 10 °C/min (Figure 6).

#### 2.2.5. Mechanical Testing

The mechanical properties and dimensional accuracy of the PCPC parts were tested using a Byes-3003 universal testing machine (Shanghai Bangyi Precision Measuring Instrument Co., Ltd. China country) and Vernier caliper (Industrial Grade IP67 Waterproof Digital. China). Tensile strength was measured in a tensile sample (150 × 20 × 10 mm) according to the official Chinese standard (GB/T1040-1992) [8,25] where a crosshead speed of 5 mm/min and gauge length of 50 mm was used. The static tensile load was determined according to the properties and dimensions of the machine test, including the total length (150 ± 0.5 mm), middle parallel part (60 ± 0.5 mm), width of the identical central part (10 ± 0.2 mm), end width (20 ± 0.2 mm), gauge length (50 ± 0.5 mm), distance between the clamps (115 ± 5 mm), radius (60 ± 0.5 mm), and thickness (4 ± 0.2 mm). The tensile strength was calculated using Equation (1):(1)$$\delta_{t}=\frac{p}{bd},$$
where *δ_t_* is tensile strength (MPa), *p* is the maximum load (N), *b* is sample width (mm), and *d* is sample thickness (mm).

Bending strength was measured in a bending sample (80 mm × 13 mm × 4 mm) according to the official Chinese standard (GB/T9341-2008 standard) [8,25], where three-point loading was applied with a crosshead speed of 0.1 mm/min and span length of 80 mm. The bending strength was calculated using Equation (2):(2)$$\delta_{f}=\frac{3FL}{2bh^{2}},$$
where *δ_f_* is bending strength (MPa), *F* is the force applied (N), *L* is sample length (mm), *b* is sample width (mm), and *h* is sample thickness (mm). The tensile and bending strength tests were conducted in triplicate and the results are presented in Table 1. Further statistical analyses of the mean values were calculated at a 95% confidence interval; furthermore, the standard deviation of the mean was calculated.

#### 2.2.6. Density

The density of the sintered PCPC parts was calculated based on the mass and dimensions of a PCPC part measured using an electronic balance and a Vernier caliper, respectively. Density analysis was applied to the measured dimensions (80 mm × 13 mm × 4 mm) and weights listed in Table 2, where the density of the sintered part was calculated using Equation (3):(3)$$d=\frac{g}{abc}$$
where *d* is density (g/cm^3^), *a* is length (mm), *b* is the width (mm), *c* is the thickness (mm), and *g* is mass (g).

#### 2.2.7. Dimensional Precision (DP)

Dimensional accuracy analysis was conducted on a sintered PCPC part with specified dimensions of 80 mm × 13 mm × 4 mm, where the actual dimensions were measured using a Vernier caliper. The dimensional accuracy was determined using Equation (4):(4)$$\delta (\%)=(1-\frac{\left[Lo-L\right]}{Lo}),$$
where *δ* (%) is the dimensional precision (%), *L* is the measured dimension (mm), and *L_O_* is the specified dimension (mm). The dimensional accuracies in the X, Y, and Z directions are presented in Table 2.

## 3. Results and Discussion

### 3.1. Single-Layer Sintering

The excellent formability of the PCPC parts was demonstrated in single-layer sintering tests, which was attributed to the bonding interface between the PCP and PES particles (Figure 7). Further, mechanical tests were used to determine the optimal powder mixture ratio. Parts produced using three different PCPC mixture ratios (10/90, 15/85, and 20/80 (wt/wt)) were evaluated under fixed processing parameters. The best sintering quality was achieved using a PCP/PES ratio of 10/90 and was found to be superior to pure PES powder.

### 3.2. PCPC Morphology

The morphology of the PCP, PES powder, and PCPC parts were evaluated by using a FEI Quanta 200 SEM scanning electron microscope (Figure 8). The PCP particles had an irregular shape and rough surface (Figure 5b), while the PES powder particles were flatter and smoother (Figure 5a). However, the PCP particle size was non-uniform. The PCP and PES particles in the 10/90 PCPC part were uniformly distributed and no agglomeration was observed (Figure 8b); the bonding interface and sintering neck between the PCP and PES is better than shown in Figure 8a.

### 3.3. Selective Laser Sintering Experiment

SLS was used to produce PCPC sintered parts with good forming accuracy. Further, the parts exhibited smooth brown surfaces (Figure 9). The sintered parts exhibited good mechanical properties, thus demonstrating the applicability of this approach to AM technology in the manufacturing of furniture, wooden floors, and construction components. The tensile, bending, and waxing test samples are shown in Figure 9a,b.

### 3.4. Mechanical Properties

The evaluation of the mechanical properties of SLS products is essential for assessing their forming quality and performance. The density and mechanical strength of the parts produced using PCPC with different mixture ratios and pure PES are shown in Figure 10. An increase in PCP content initially led to an increase in bending and tensile strength of the sintered parts, where maxima of 10.78 and 4.94 MPa, respectively, were achieved at 10% PCP (10/90 wt/wt). Thereafter, the mechanical strength decreased. PCP enhanced the sintering quality of the parts, which caused complete melting and mixing of the PES. This led to firm cohesion of the PES powder and PCP, which improved the density and mechanical strength of the sintered PCPC parts. Pure PES is often over decomposed during sintering in SLS, which causes a reduction in sintering quality. However, the PCP absorbed a large proportion of the laser energy and minimized this damage. Further, the smaller proportion of PES led to a reduction in cost. The density and mechanical strength of the sintered PCPC parts at a PCP/PES ratio of 10/90 were presented with a best value in bending and tensile strength. In addition, these parts exhibited high dimensional accuracy and were better than pure PES (Table 1). The dimensional accuracy of the pure PES sintered part is 99.88, 99.42, and 91.015 in X, Y, and Z direction, respectively. This result is lower than 10/90 when compared to the data in Table 1. However, the loss in density and mechanical strength at higher PCP contents was attributed to the lower proportion of the PES powder binder, where a decrease in interface bonding strength between the PCP and PES powders was observed. Moreover, the internal holes between the PCP and PES powder was increased in a high content of PCP (Figure 8a). The bending and tensile strengths of the parts are presented in Figure 11. Further statistical analyses of the mean values were calculated at a 95% confidence interval. Furthermore, the standard deviation of each sample test was calculated. All of the mechanical strength measurements fell within the upper and lower limits of the 95% confidence interval, demonstrating the reliability of the results. The mechanical strength and surface roughness of the sintered PCPC parts were further improved using post-processing, specifically via infiltration with wax. The majority of the inner pores were filled with wax, thus reducing the void fraction between the PCP and PES powder. The mechanical properties of the PCPC-waxed parts were enhanced due to the increased density, where the bending strength and tensile strength after wax infiltration were enhanced from 10.78 to 12.38 MPa and 4.94 to 5.73 MPa. Thus, the bending strength and tensile strength after wax infiltration improved by 13% and 14%. Further, the surface roughness quality was improved with a reduced value from 6.87 μm to 4.886 μm. The laser sintering experiments and mechanical testing demonstrated that a PCP/PES ratio of 10/90 PCPC was optimal (Figure 7 and Figure 10), which is recommended to be used in future research. The mechanical properties of the PCPC were much higher than previous reports on walnut shell/co-PES and wood–plastic composites by Yueqiang and Zeng (Table 2). SLS parts produced using a sisal fiber/PES composite had a higher mechanical strength than PCPC; however, PCPC is lower in cost and has more economical and environmental benefits.

### 3.5. The Surface Quality of the Sintered Part

The surface quality of sintered parts is an important characteristic. The differences in the surface forming quality of the parts before and after post-processing, and after polishing the wax infiltration part were evaluated (Figure 12). The single-layer sintering experiments and SEM confirmed that 10% PCP was optimal (Figure 7). The SEM observation of the 20% PCP part revealed the formation of many large internal holes due to insufficient melting of the PES particles, which caused a reduction in the sintering neck of the PCPC (Figure 8a). The high content of PCP hindered sufficient irradiation of the laser energy on the PES particles, which led to less melting. The small amount of molten PES powder was unable to bond and surround the large number of PCP particles, which led to a reduction in the quality of the PCPC part. The surface morphology of the 10/90 PCPC part was characterized by a smaller quantity and smaller size of the internal holes and a larger sintering neck (Figure 8b). These characteristics led to superior density, mechanical properties, and forming quality (Figure 10 and Figure 11).

### 3.6. Sintering Temperature of the PCPC

SLS technology is a method that mainly depends on thermal influences. However, PCP has no melting point, and PES is an amorphous polymer; thus, the PES powder plays a significant role in the formation of the PCPC composite. The glass transition temperature of PES and PCP were found through the DSC test, and then the SLS processing (preheating temperatures) were determined accordingly to the DSC result curve. To prevent the PCPC sintered parts from warping in the process of sintering, the powder material was preheated within a specific temperature range based on the DSC test [25]. The temperature range represents the sintering window, as presented in Ts and Tc. Ts is the softening point, while Tc is the caking temperature. PES is a non-crystallizable polymer; Ts is the glass transition temperature (Tg) of PES and PCP powders. However, Tc cannot be determined from the DSC curves, but it is observed through the experiment. The DSC curves of the PES and PCP powders are shown in Figure 6, where Tg of PES and PCP should be not more than (88 and 89 °C) based on the DSC test. Through the experiments, it can be observed that the PES and PCP powders can be completely caking at 114 °C and 116 °C, respectively. Consequently, the sintering windows and the glass transition temperature (Tg) of the PES and PCP powders are PES ≤ 88 °C and PCP ≤ 89 °C, respectively. Note: The preheating temperature of PES and PCP in this study should be not more than 88 °C and 89 °C, this means the Tg of the PES and PCP powder are less than or equal to 88 °C and 89 °C. To produce the PCPC parts, the preheating temperatures were set within temperature ranges. Consequently, the preheating temperature of the PCPC composite was selected within a temperature range (sintering windows), which was 74 °C, 78 °C, 82 °C, and 86 °C.

## 4. Conclusions

In this study, a novel sustainable composite material of PCP and PES powder was developed as an SLS material. This research aimed to extend the range of available materials of the SLS process. High quality parts were produced by optimizing the mixture ratio of the raw materials and post-processing. Firstly, the addition of PCP led to an increase in mechanical strength of the SLS parts and then gradually decreased. The quantity and size of the internal holes within the sintered parts were lowest with a small proportion of PCP (10%), but these gradually increased as the content increased, leading to a decrease in the density and mechanical strength of the sintered part. The bending strength (10.78 MPa) and tensile strength (4.94 MPa) of the SLS part with a PCP/PES mixture ratio of 10/90 (wt/wt) exhibited the highest mechanical strength and better forming quality than the 15/85 and 20/80 PCPC and was superior to pure PES. Further, the dimensional accuracy of the 10/90 PCPC parts was excellent in all three dimensions (X, Y, and Z directions); and better than pure PES. The post-processing treatment resulted in a substantial improvement of the mechanical properties and surface roughness of the sintered PCPC parts, where the bending and tensile strengths of the wax-infiltrated parts increased from 10.78 to 12.38 MPa and 4.94 to 5.73 MPa, respectively, where the rate of increase was equivalent 13% and 14% for bending and tensile, respectively. Further, the surface roughness value was significantly reduced after post-processing to an optimal level from 6.87 to 4.886 μm. The mechanical properties of PCPC parts showed better results in the SLS test than the ones presented by Guo et al. [12] and Yueqiang [13].

## Figures and Tables

**Figure 1 materials-13-03034-f001:**
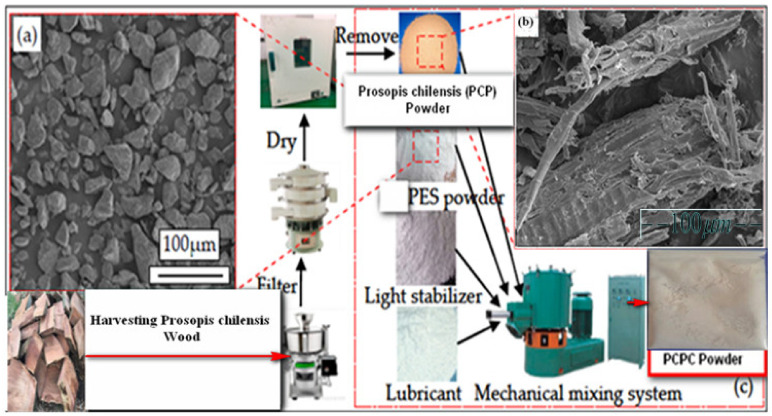
Preparation of the Prosopis chilensis/polyethersulfone composite (PCPC), with SEM micrographs of the (**a**) polyethersulfone (PES) powder particle morphology, (**b**) Prosopis chilensis powder (PCP) particle morphology, and a digital photograph of the (**c**) PCPC powder after mechanical mixing.

**Figure 2 materials-13-03034-f002:**
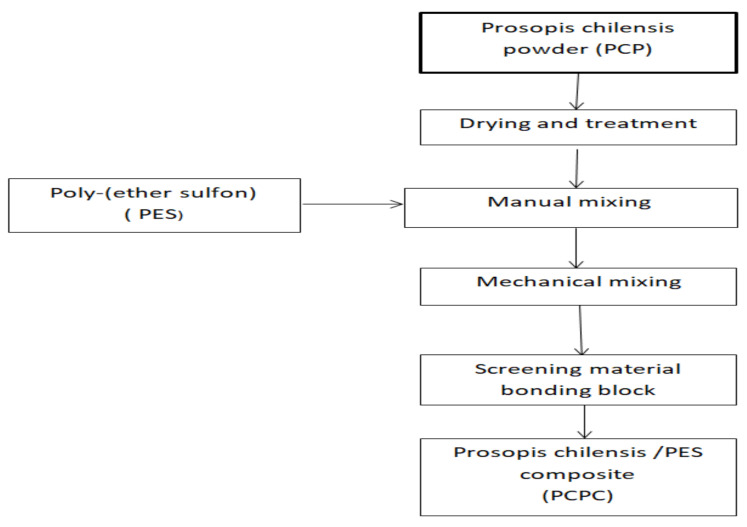
Flow chart of the preparation process of Prosopis chilensis/PES composites (PCPC).

**Figure 3 materials-13-03034-f003:**
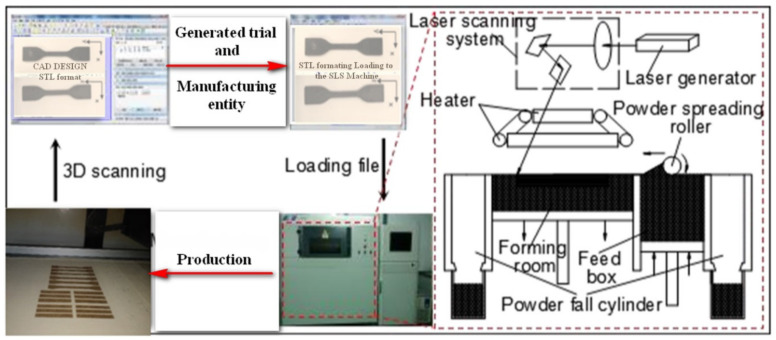
Characteristic building process using a AFS-360 rapid prototyping machine.

**Figure 4 materials-13-03034-f004:**
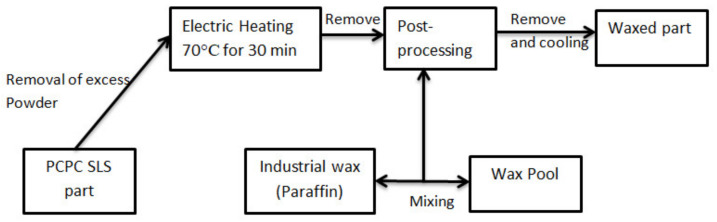
The post-processing flow chart of the PCPC parts.

**Figure 5 materials-13-03034-f005:**
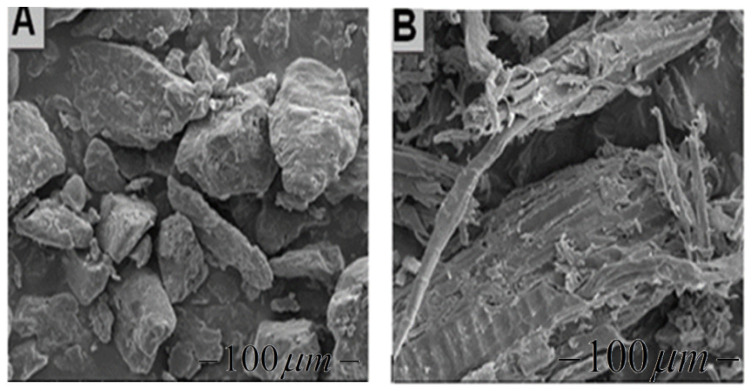
Surface morphology of (**A**) the PES powder and (**B**) the PCP.

**Figure 6 materials-13-03034-f006:**
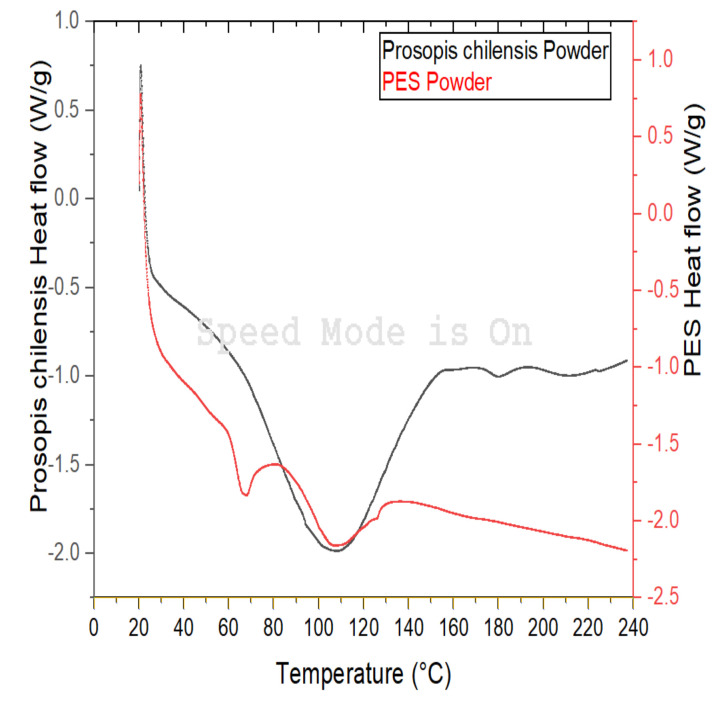
DSC curve of the PES and PCP powders between 20 °C and 240 °C at a heating rate of 10 °C/min.

**Figure 7 materials-13-03034-f007:**
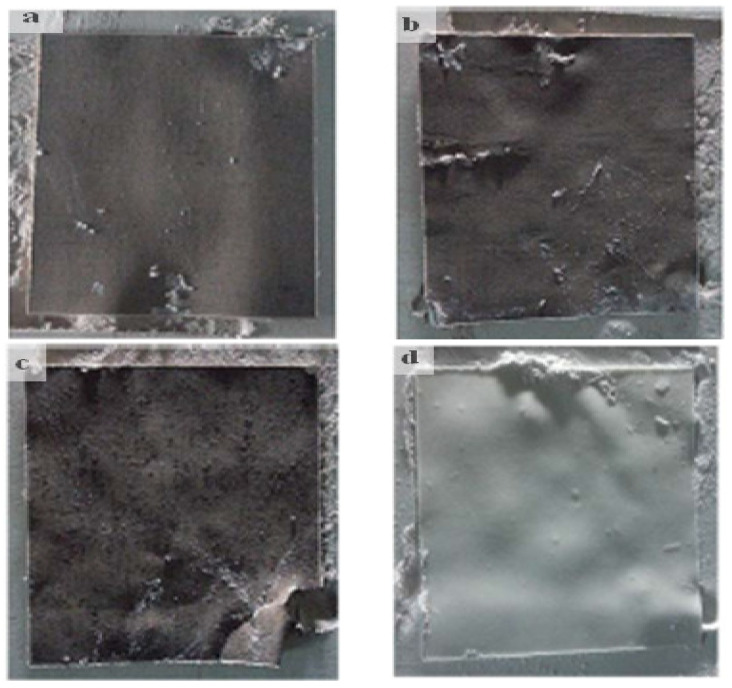
Single-layer sintered PCPC and PES samples prepared using different PCP and PES powder ratios, namely, (**a**) 10% PCP (10/90), (**b**) 15% PCP (15/85), (**c**) 20% PCP (20/80), and (**d**) pure PES.

**Figure 8 materials-13-03034-f008:**
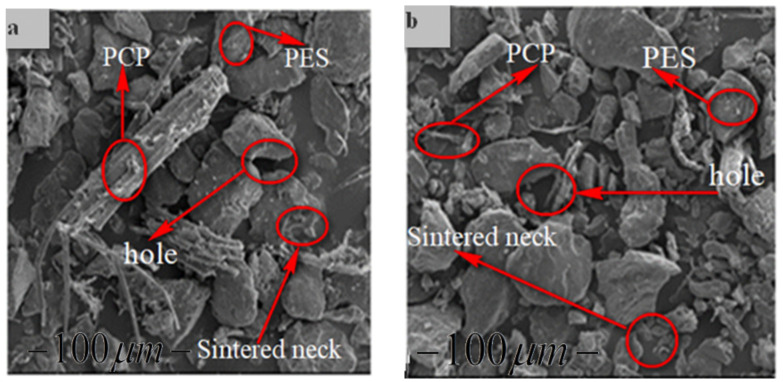
Surface morphology of (**a**) PCPC parts (20% PCP) and (**b**) PCPC parts (10% PCP).

**Figure 9 materials-13-03034-f009:**
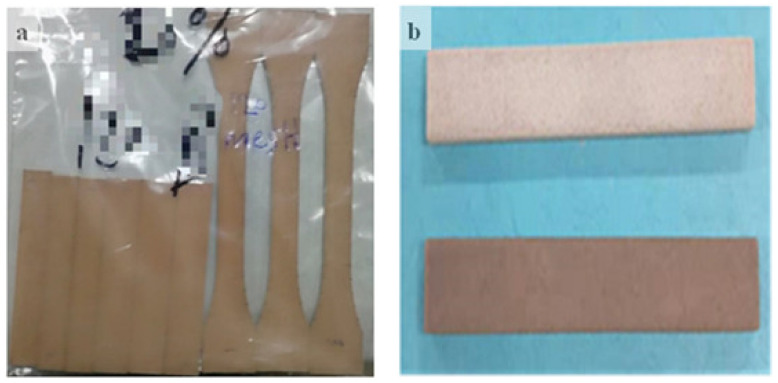
Selective laser sintering (SLS) samples for (**a**) the tensile and bending tests, and (**b**) the waxing test.

**Figure 10 materials-13-03034-f010:**
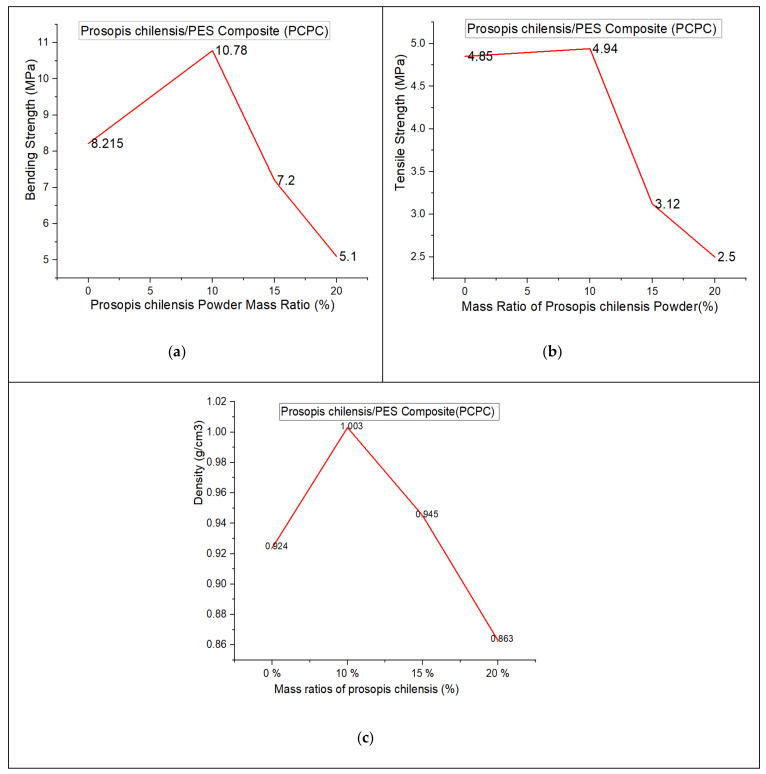
Mechanical properties of the PCPC parts with varying mixture ratios, namely, (**a**) bending strength, (**b**) tensile strength, and (**c**) density. (Note: zero mass ratio represents a pure PES sintered part).

**Figure 11 materials-13-03034-f011:**
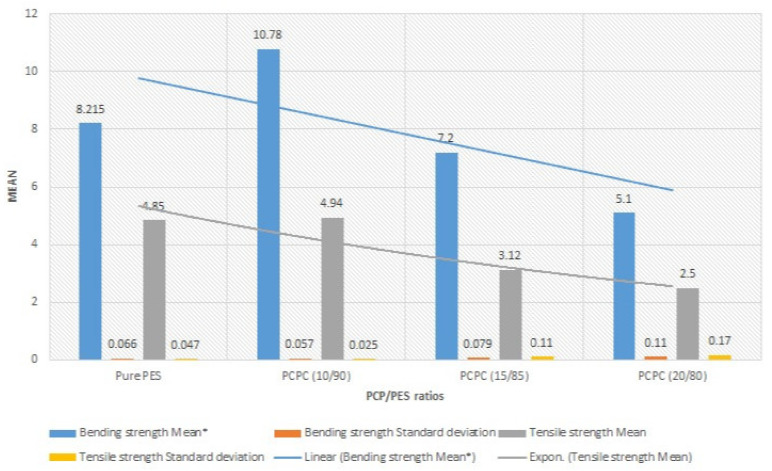
Mechanical properties of the PCPC parts with varying PCP/PES ratios including the standard deviation of the mean. The lower and upper limits of the bending tests were (7.22, 9.22), (10.13, 11.43), (6.55, 7.85), and (4.45, 5.75), respectively, from pure PES to PCPC (20/80); and the lower and upper limits of the tensile tests parts were (4.27, 5.43), (4.30, 5.59), (2.47, 3.78), and (1.85, 3.15), respectively, from pure PES to PCPC (20/80). Note: the lower and upper limits of the bending and tensile strength are the 95% confidence intervals for the means. *: means any mean value in chart became between the upper and lower values.

**Figure 12 materials-13-03034-f012:**
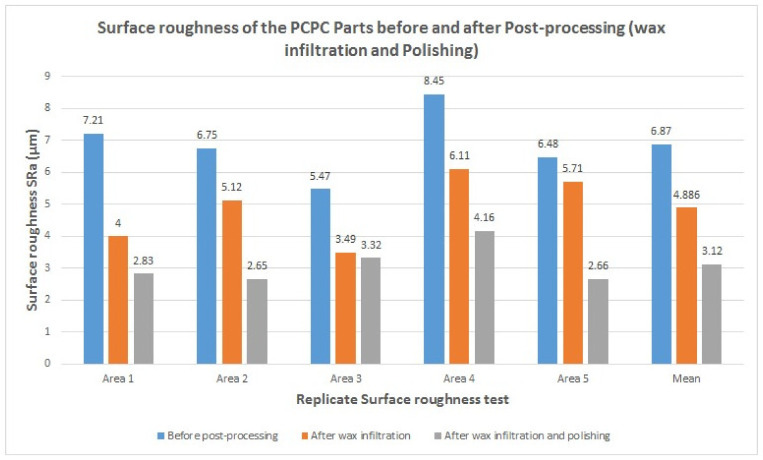
Surface roughness of the PCPC parts before and after post-processing (wax infiltration and polishing).

**Table 1 materials-13-03034-t001:** The density and dimensional accuracy of the pure PES and PCPC parts with varying PCP/PES ratios.

PCP/PES Ratios	Density (g cm^3^)	Dimensional Accuracy of the PCPC Parts (%)		
		X	Y	Z
Pure PES	0.924	99.88	99.42	91.015
PCPC (10/90)	1.003	99.937	99.572	92.024
PCPC (15/85)	0.945	99.883	99.40	90.703
PCPC (20/80)	0.863	99.862	99.380	88.495

**Table 2 materials-13-03034-t002:** Comparison of the mechanical properties composite materials produced under the same conditions.

Material	Tensile Strength (MPa)	Bending Strength (MPa)	Reference
Prosopis chilensis/PES (20/80 wt/wt)	2.50	5.10	Current study
Wood/plastic	2.17	3.22	Guo et al. 2011 [12]
Walnut shell/co-PES (20/80 wt/wt)	2.3	4.2	Yueqiang et al. 2017 [13]
Sisal fiber/PES (SFPC) (20/80 wt/wt)	3.24	7.12	Li Jian et al. 2020 [8]

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
