# Peer review of "Selective Laser Sintering (SLS) and Post-Processing of Prosopis Chilensis/Polyethersulfone Composite (PCPC)"

_materials, 2020, doi:10.3390/ma13133034_

Round 1

Reviewer 1 Report

Major revision is necessary. Comments are below:

Results are not properly discussed.

Comments are below:

1) page 3 102 line

The molecular structure of PES

102  includes three groups, namely a sulfone, ether, and phenylene group.

Details of the used PES should be provided.

2) line 105

"The glass transition temperature (Tg) of PES and PCP"; Please comment about PCP  Tg?

3) Fig.1 and 2 quality need to be enhanced. These figures are not properly discussed in the text.

4) Voids should be evaluated before and after post processing

5) Fig. 4 ; Scale is necessary

6) Fig 5. need to be discussed properly.

7) Fig. 6 , Fig 7 Scale is necessary

8) The measurements errors need to be provided.

9) Table 1 ; why several data are in BOLD

10) Why Fig 10 is reported? It is not discussed.

11) Abstract and conclusions section should be strongly enhanced.

Author Response

Responses to reviewers,

We would like to thank the editors and reviewers for giving as the second round an opportunity to revise our manuscript, and for their constructive comments and suggestions on our manuscript. We have addressed the major concerns of the reviewers. More specifically, we have also included a point-by-point response to the reviewers, see replies to the reviewers' comments below for details.

Reviewer 1 comment:

Major revision is necessary. Comments are below:

Results are not properly discussed.

Comments are below:

1) Page 3 102 line. The molecular structure of PES 102 includes three groups, namely a sulfone, ether, and phenylene group. Details of the used PES should be provided.

Response : Thank you very much for your good observations and evaluation, which helps us to improve the manuscript in our future work, and we are very sorry for our incorrect or missing some essential of various parts of the manuscript. Answer 1: According to your comment we presented the details of the PES powder. PES is a type of thermoplastic polymer with excellent comprehensive properties and a stable performance temperature range (-100 °C to 200 °C), which was developed by the British Imperial Chemical Industry Group in the 1970s (Knoell et al. 1999). PES is one of the most widely used materials for special engineering plastics. The molecular structure of PES is primarily composed of three groups, the ether groups, sulfone groups, and a phenylene group (Knoell et al. 1999). Please see Lines (102- 106).

2) Line 105 "The glass transition temperature (Tg) of PES and PCP"; please comment about PCP Tg?

Response 2: We would like to thank you in advance for your positive comments. Answer 2: We revised the manuscript according to your comments. SLS technology is a method that mainly depends on thermal influences. However, PCP has no melting point, and PES is an amorphous polymer; thus, the PES powder plays a significant role in the formation of PCPC composite. The glass transition temperature of PES and PCP were found through the DSC test, and then the SLS processing and preheating temperatures were determined accordingly to DSC result curve. To prevent the PCPC sintered parts from warping in the process of sintering, the powder material was preheated within a specific temperature range based on the DSC test. Where, Tg of PES and PCP should be not more than (88 and 89°C ) . Please see Lines (109- 112). For more details pleas go to DSC test discussion in lines 286-303.

3) Fig.1 and 2 quality need to be enhanced. These figures are not properly discussed in the text.

Answer 3: We revised the manuscript according to your comments; moreover, we added a new flow chart to more explain the process in Fig.1. Please see lines (114-116, and 119-121).

4) Voids should be evaluated before and after post processing.

Answer 4: The voids between the particles before and after post processing were evaluated through surface roughness test; where, the number of voids was reduced after post processing , besides that the density and mechanical strength were significant improved after post processing, this enhance refer to reduce the voids friction between the PES and PCP particles. The surface roughness after post processing improved from 6.87 μm to 4.886 μm.

5) Fig. 4; Scale is necessary.                                        

Answer 5: According to your comments we revised all the Figs scale in manuscript. Please see lines (147-149 and 214-215). Based on other reviwer comment we divided Fig. 4 into two Figs and the new is (Fig. 5 and Fig. 8)

6) Fig 5. need to be discussed properly.          

Response: thank you very much for your evaluation, which helps us to improve future work. Answer 6: Firstly Fig 5 now become Fig 6. We are very sorry for missing the discussion of Fig 5. SLS technology is a method that mainly depends on thermal influences. However, PCP has no melting point, and PES is an amorphous polymer; thus, the PES powder plays a significant role in the formation of PCPC composite. The glass transition temperature of PES and PCP were found through the DSC test, and then the SLS processing (preheating temperatures) were determined accordingly to DSC result curve. To prevent the PCPC sintered parts from warping in the process of sintering, the powder material was preheated within a specific temperature range based on the DSC test. The temperature range represents the sintering window, which, as presented in (Ts and Tc). Ts is the softening point, while Tc is the caking temperature. PES is a non-crystallizable polymer; Ts is the glass transition temperature (Tg) of PES and PCP powders. However, Tc cannot found from the DSC curves, but it observed through the experiment. The DSC curves of the PES and PCP powders are shown in Fig. 6, where Tg of PES and PCP should be not more than (88 and 89°C) based on DSC test. Through the experiments, it can be observed that the PES and PCP powders can be completely caking at (114°C and 116°C), respectively. Consequently, the sintering windows and the glass transition temperature (Tg) of the PES and PCP powders are: PES ≤ 88°C and PCP≤ 89°C respectively. Note. The preheating temperature of PES and PCP in this study should be not more than 88°C and 89°C, this means the Tg of the PES and PCP powder are less than or equal (88°C and 89°C). To produce the PCPC parts, the preheating temperatures was set within temperature ranges. Consequently, the preheating temperature of the PCPC composite was selected within a temperature range (sintering windows), which as (74°C, 78°C, 82°C, and 86°C). Please see the discussion of Fig 6. in lines (288-305). The urgent need to develop new natural and environmentally-friendly materials, with low energy consumption, low CO2 emission and low cost.

7) Fig. 6, Fig 7 Scale is necessary.                  

Answer 7. According to your comments we revised all the Figs scale in the manuscript. Please see lines (201-203 and 223-224) in Figs 7 and 9.                                      

8) The measurements errors need to be provided.

Answer 8. The measurements errors represent the value between the standard value and actual measured value. The reference standard given by the SLS machine in the bending part test is (80.03 mm × 12.95 mm × 4.3 mm); this value represents the measured (actual) value. However, the standard value in STL file is (80 mm × 13 mm × 4 mm), for length, width and thickness, respectively. For example, the error in (X) direction= 0.03mm, (Y) direction=0.05mm, and (Z) direction=0.3mm.

9) Table 1; why several data are in BOLD.

Answer 9. We very sorry for that. We revised the manuscript according to your comments. Please see details in Table 1 in lines (277-278).

10) Why Fig 10 is reported? It is not discussed.                      

Answer 10. We are very sorry for reported Fig 10; we revised please see the Fig. 10 in line 260.

11) Abstract and conclusions section should be strongly enhanced.

Response: thank you very much for your evaluation. Answer 11. Based on your comment we revised the manuscript and re-discussion the abstract and conclusions section. Please see the modified version in lines ( and 14-28, 307-323).

Reviewer 2 Report

  1. Please provide some practical examples regarding the usage of the STS materials.
  2. Which are the mechanical properties of the PES polymer?
  3. In Figure 1, the PCPC mixture structure (picture c) should be better higlighted, with a bigger image.
  4. In lines 116 – 119: how have been established the main processing parameters?
  5. Same question for figure 3.
  6. Regarding equation (1): the density was calculated for 100% brick-shaped parts? Can you provide a picture about these parts?
  7. For equations (2) and (3) it would be nice to ilustrate with some pictures the loading schemes.
  8. How was established equation (3)? The bending stress is determined as the ratio between the bending moment (F*L) and the section modulus (b*h*h/6). Why the relatio (3) looks different?
  9. For the conclusions: please provide a comparativelly study of the obtained results with other results obtained in the references.

Author Response

Responses to reviewers,

We would like to thank the editors and reviewers for giving as the second round an opportunity to revise our manuscript, and for their constructive comments and suggestions on our manuscript. We have addressed the major concerns of the reviewers. More specifically, we have also included a point-by-point response to the reviewers, see replies to the reviewers' comments below for details.

Reviewer 2 comment:

Comments and Suggestions for Authors

  1. Please provide some practical examples regarding the usage of the STS materials.

Answer 1: The expected potential environmental benefits of PCPC product is reducing the influence of pressure on the forest sector in Sudan that is cut to use in home furniture purpose,,. Prosopis chilensis composite used for parquet floors, doors, furniture, construction and other purpose. The Popular Applications for Polyethersulfone (PES) are expanded from aircraft interiors, automotive and electrical applications to medical and consumer uses, today PES has several market applications. PES Replaces Metals/ Thermosets in Automotive. PES has found growing usage in automotive applications. PESU resin reinforced with glass or carbon fibers is increasingly used in automotive components & thus replacing metal and thermoset materials. Polyether sulfones are very high temperature resistant amorphous thermoplastics that are used where the performance requirements exceed the capabilities of other engineering plastics. Please see lines (75-78).

  1. Which are the mechanical properties of the PES polymer?

Response: Thank you very much for your good observations and evaluation, which helps us to improve the manuscript in our future work. Answer 2: PES is a type of thermoplastic polymer with excellent comprehensive properties and a stable performance temperature range (-100 °C to 200 °C), (Knoell et al. 1999). PES is one of the most widely used materials for special engineering plastics. The molecular structure of PES is primarily composed of three groups, the ether groups, sulfone groups, and a phenylene group (Knoell et al. 1999). The mechanical properties of the PES are: PES has higher mechanical properties and aging resistance than most thermoplastic resins; it can be easily processed and exhibits low mold shrinkage, the ability of PES to endure repeated sterilization allows it to be used in a variety of medical applications. Very good hydrolytic and sterilization resistance. Biocompatibility, excellent insulation properties. Besides, an amorphous polymer which possesses bonds of high thermal and oxidative stability, Sulfone group provides high temperature performance, Ether linkage contributes toward practical processing by allowing mobility of the polymer chain when in the melt phase.

  1. In Figure 1, the PCPC mixture structure (picture c) should be better higlighted, with a bigger image.

Response. We would like to thank you in advance for your positive comments/suggestions. Answer 3: We revised the manuscript according to your comments; moreover, we added a new flow chart to more explain the process in Fig.1. Please see lines (114-116, and 119-121).

  1. In lines 116 – 119: how have been established the main processing parameters?

Response: Thank you very much for the positive comment. Answer 4: In the principle of SLS, the heat of composite material not more than the glass transition of the materials (Tg).Through the several single-layer sintering experiments and DSC test we estimated the suitable processing parameters of SLS part of PCPC. Moreover, the urgent need to develop new natural and environmentally-friendly materials, with low energy consumption, low CO2 emission and low cost for SLS.

  1. Same question for figure 3.

Answer 5: First of all we modified the figure 3 at a better form and now become Fig. 4. We selected the industrial paraffin and wax pool materials in the processing infiltration method because both materials have property of capillarity action within the voids between the particles. Besides that this materials were used in post-processing infiltration with wax in various previously studies and had good result. (Li, J., 2020).

  1. Regarding equation (1): the density was calculated for 100% brick-shaped parts? Can you provide a picture about these parts?

Answer 6: The density of the sintered PCPC parts was calculated based on the mass and dimensions of a PCPC part measured using an electronic balance (b) and Vernier caliper(a), respectively. Density analysis was applied to the measured dimensions (80 × 13 × 4 mm); and the picture below (c) shown the PCPC part used in density measurement analysis.

   (a) Vernier caliper (device used to determine the dimensions of X, Y, and Z direction).

   (b) High sensitive balance scale.         (C) SLS bending part that used in density analysis.

  1. For equations (2) and (3) it would be nice to illustrate with some pictures the loading schemes.

Response: Thank you very much for your good observations. Answer 7. Surely, please see the images below. Fig. (a) Shown the samples of PCPC including, tensile and bending parts, (b) the universal testing machine, and (c), is shown the picture of PCPC tensile part under mechanical testing loading schemes.

(a) SLS samples for bending and tensile tests. (b) Universal testing machine, (c) SLS tensile part under mechanical testing loading schemes.                                               

  1. How was established equation (3)? The bending stress is determined as the ratio between the bending moment (F*L) and the section modulus (b*h*h/6). Why the relatio (3) looks different?

Answer 8: We determined the bending strength according to this equation; this equation have relation between the force (f) , length (L) , width (b), and thickness of SLS part (h). And also we cited more papers in term of this issue (Li, J., 2020a, and Idriss, Aboubaker. 2020).We thing there is no difference, but here the shape is a rectangle. Thus, the moment inertia (I) is varies from shape to other. Besides that the bending and tensile strength was calculated with using universal testing machine (b).

where is bending strength (MPa), F is the force applied (N), L is sample length (mm), b is sample width (mm), and h is sample thickness (mm).

  1. For the conclusions: please provide a comparativelly study of the obtained results with other results obtained in the references.

Answer 9: As you recommended we revised the conclusions section and showed the aim of this study; besides, we were compared it with other results. Please see lines (307-323).

Reviewer 3 Report

The paper presents the laser-sintering fabrication of the Prosopis Chilensis with the polyethersulfone. After the deposition process, the post-treatment was conducted. The idea of the paper is interesting. It meets with the current "ecological way of thinking" in materials engineering. I have the following comments on the paper:

  • in Keywords: Please remove "and" word; also "PES powder" is not clear
  • In the introduction, please discuss the post-treated methods of the material-type which is the object of your study. Present the processing methods given by the literature.
  • I think that the Appendix should be added. Please put there the material from fig. 9 or simply delete it.
  • Fig. 4 should be divided into two figs. The C and D figs should be placed in the "results section", where authors describing the result of SEM. Or all SEM should be placed in section 3.2. Also please mark "polymer" and" wood" particles in figs C and D. Moreover, the scale bar is needed in all SEM microphotos.
  • Explain why you did not present the mechanical properties of post-treated samples? I think it should be given. The results should be presented and discussed with the relation to untreated samples. in conclusion, the authors mention: "where
    288 the tensile and bending strengths of the wax-infiltrated parts increased by 14% and 13%, respectively" but the discussion does not contain any raw data for infiltrated components. 
  • Please add a scale bar in fig. 6. 
  • Discussion of the findings with the literature data is not sufficient. I think that the discussion of the obtained results with the literature data should be developed and commented. 
  • Please improve the style of the phrase "The density and mechanical strength of the sintered PCPC parts at
    229 a PCP/PES ratio of 10/90 were excellent." (excellent?) 
  • In the phrase "The dimensional accuracy of the pure PES sintered part is (99.88, 99.42,
    231 and 91.015 in X, Y, and Z direction, respectively)" please refer to the results of specific tab or fig.
  • The authors wrote: "the internal holes and a larger sintering neck (Fig. 4d)." Please mark them in fig 4.
  • I think that fig 8 should be transferred to section 3.4. It contains the mechanical results - which are discussed in the previous section. 
  • I suggest to preset the table 1 in a graph (plot) form. Now the results are difficult to read.
  • Table 2 mostly, duplicates the information given in fig.8 therefore only the "Accuracy" results should be left and presented in a graph (plot form). 
  • In table 3 - references numbering should be given [x].
  • In tab3., - It is not clear why only 80/20 literature results are compared with your 80/20 findings? please explain why you compare it in that way while according to your findings, "the best sample" was 90/10.
  • Table 4 - information should be presented in a column graph form.
  • Conclusions - are not clear. In the beginning, the object and the aim of the study should be shortly characterized. Now, "initial increase" and "higher content" phrases are not clear.
  • Also, the phrase "tensile and bending strengths of the wax-infiltrated parts increased by 14% and 13%" does not derive from the text.
  • References style should be improved according to the journal requirements. 

Author Response

Responses to reviewers,

We would like to thank the editors and reviewers for giving as the second round an opportunity to revise our manuscript, and for their constructive comments and suggestions on our manuscript. We have addressed the major concerns of the reviewers. More specifically, we have also included a point-by-point response to the reviewers, see replies to the reviewers' comments below for details.

Reviewer 3 comment:

The paper presents the laser-sintering fabrication of the Prosopis Chilensis with the polyethersulfone. After the deposition process, the post-treatment was conducted. The idea of the paper is interesting. It meets with the current "ecological way of thinking" in materials engineering. I have the following comments on the paper: Response: thank you very much for your good observations and evaluation, which helps us to improve the manuscript in our future work, and we are very sorry for our incorrect or missing some essential of various parts of the manuscript.

  1. In Keywords: Please remove "and" word; also "PES powder" is not clear

Answer 1. We revised the keywords of manuscript according to your comments. Please see in lines (29-30).

  1. In the introduction, please discuss the post-treated methods of the material-type which is the object of your study. Present the processing methods given by the literature.

Response: We would like to thank you in advance for your positive comments. Answer 2. We discussed the post-treated method in the introduction section. Please see in Lines (84-88).

  1. I think that the Appendix should be added. Please put there the material from fig. 9 or simply delete it.

Answer 3. According to your comment/suggestions we deleted the Fig. 9 from the manuscript.

  1. Fig. 4 should be divided into two figs. The C and D figs should be placed in the "results section", where authors describing the result of SEM. Or all SEM should be placed in section 3.2. Also please mark "polymer" and" wood" particles in figs C and D. Moreover, the scale bar is needed in all SEM microphotos.

Answer 4. As your comment, we divided Fig. 4. Into two figs and was transferred the Fig. 4 (c and d) to result section (3.2) and marked the Figs C and D to explain the PES, PCP, and holes and sintering neck between particles. Please see lines (147-148 and 214-215). Fig.4. now become Fig. 5.

  1. Explain why you did not present the mechanical properties of post-treated samples? I think it should be given. The results should be presented and discussed with the relation to untreated samples. In conclusion, the authors mention: "where 288the tensile and bending strengths of the wax-infiltrated parts increased by 14% and 13%, respectively" but the discussion does not contain any raw data for infiltrated components.

Response: Thank you very much for your positive comment. Answer 5. According to your comment, we were refer to the post-processing of PCPC part in introduction section; besides, we evaluated the mechanical strength after post-processing and was mentioned in discussion section. The bending and tensile strength and surface roughness of the sintered PCPC parts were improved after post-processing. Where the bending strength and tensile strength after wax infiltration were enhanced from 10.78 to 12.38 MPa and 4.94 to 5.73 MPa, respectively. Further, the surface roughness quality was improved with a reduced value from 6.87 μm to 4.886 μm. Therefore, tensile and bending strengths after wax-infiltrated increased by 14% and 13%, respectively; that mean the bending strength after wax infiltration was enhanced from (10.78 to 12.38 MPa, where the increase rate is 14%) and tensile strength was enhanced from (4.94 to 5.73 MPa, where the increase rate is 13%). Please see details in lines (84-88 and 248-254).

  1. Please add a scale bar in fig 6.

Answer 6. According to your comment we revised all the Figs scale in manuscript. Please see details in lines (201-203).

  1. Discussion of the findings with the literature data is not sufficient. I think that the discussion of the obtained results with the literature data should be developed and commented.

Response: thank you very much for your good observations and evaluation, which helps us to improve the manuscript in our future work. Answer 7. Based on your comment, we revised the discussion of the results by adding unique paragraph (Sintering temperature of the PCPC. (Discussion Fig. 6)), besides, enhanced the conclusion and abstract section. Please see in Lines (288-305 and 307-323).

  1. Please improve the style of the phrase "The density and mechanical strength of the sintered PCPC parts at 229 a PCP/PES ratio of 10/90 was excellent." (excellent?)

Response: Thank you very much for your good comment, and we are very sorry for our incorrect writing in some parts of the manuscript. Answer 8. As your comment, we are refining writing to the whole text of our revised manuscript, and we tried to avoid any grammatical or syntax errors. Please see Lines (237-238).                      

  1. In the phrase "The dimensional accuracy of the pure PES sintered part is (99.88, 99.42,and 91.015 in X, Y, and Z direction, respectively)" please refer to the results of specific tab or Fig.

Response: thank you very much. Answer 9. Based on your comment, we referred to the accuracy and transferred it to the results in Table 2. Please see in Lines (239-241 and 279-280).                                            

  1. The authors wrote: "the internal holes and a larger sintering neck (Fig. 4d)."Please mark them in fig 4.

Answer 10: According to your comment, we revised Fig. 4d (we were divided the Fig. 4 into two figs and was transferred the Fig. 4 (c and d) to the result in section (3.2) and was marked the C and D to explain the PES, PCP, and holes and sintering neck between particles; this work we done based on comment 4. Please see in lines (214-215). Note. Fig. 4 (c and d) now become Fig. 8 (a and b).

  1. I think that fig 8 should be transferred to section 3.4. It contains the mechanical results - which are discussed in the previous section.                                                                                                                                                                                                                                                                                                                                                                                                                          

Response: We would like to thank you in advance for your positive comments/suggestions. Answer 11:

We transferred the Fig 8 to section 3.4. Please see in lines (261-262). And now become Fig. 10.

  1. I suggest to preset the table 1 in a graph (plot) form. Now the results are difficult to read.                

Thank you very much for positive comment and encouraging suggestion, which help us to improve the sequence quality by avoiding the reduplicative in the discussion. Answer 12: we revised and redesigned all the Tables of this manuscript and try to avoid any reduplicative in the discussion section; however, in some causes it’s necessary to mention the number of replicate test and mean of result because it important to explain the 95% confidence intervals for the mean, which is shown the reliability of the results to the readers. These reasons lead me to put the Tables again. If you do not agree to this revise we are ready to modify it again. Thank you in advance. Please see the new Table in lines (277-278).

  1. Table 2 mostly, duplicates the information given in fig.8 therefore only the "Accuracy" results should be left and presented in a graph (plot form).

Thank you for your comment. Answer 13: we revised and redesigned Table 2 and removed any reduplicative information. Now, the new Table 2 just explains the dimensional accuracy of the PCPC parts according to mass ratios between PES and PCP particles. Please see in lines (279-280).                                                                                                                                                                                                                                                

  1. In table 3 - references numbering should be given [x].

Answer 14: As you recommended we revised and numbered the references in Table 3. Please see in lines (281-283).

  1. In tab3., - It is not clear why only 80/20 literature results are compared with your 80/20 findings? Please explain why you compare it in that way while according to your findings, "the best sample" was 90/10. Response: Thank you very much for your positive comment. Answer 15: In fact, the best sample in the current study is 90/10; however, we compared it with sample 80/20 because most the previous studies in Table 3 started the experiments from mixture ratios of 80/20.
  2. Table 4 - information should be presented in a column graph form.

Answer 16: We redesigned Table 4 by the flexible way and now were accessible to understanding the information and also better than column graph. Thank you very much. If you need more modification we are ready and willing to do more revise in my future work as you like. Please see the new Table 4 in lines (284-286).

  1. Conclusions - are not clear. In the beginning, the object and the aim of the study should be shortly characterized. Now, "initial increase" and "higher content" phrases are not clear.

Answer 17: As you recommended we revised the conclusions section and showed the aim of this study; besides, we were compared it with other results. Please see lines (307-323).

  1. Also, the phrase "tensile and bending strengths of the wax-infiltrated parts increased by 14% and 13%" does not derive from the text.

Answer 18: According to your comment we connected the information in all parts of the article. Please see in lines (84-88, and 248-254).

  1. References style should be improved according to the journal requirements.

Answer 19: As you recommended we revised all the references style according to the journal guidelines. Please see references section in lines (330-389).

Round 2

Reviewer 1 Report

Paper can be published

Author Response

Responses to reviewers,

We would like to thank the editors and reviewers for giving as the second round an opportunity to revise our manuscript, and for their constructive comments and suggestions on our manuscript. We have addressed the major concerns of the reviewers. More specifically, we have also included a point-by-point response to the reviewers, see replies to the reviewers' comments below for details.

Reviewer 2 comment:

Comments and Suggestions for Authors

Thank you for your responses. However, I am not satisfied before the SEM (and other) photos are not improved. Still, the scale bar (marker) should be added e.g. in the figs 5 and 8. Even the authors claim that it is given - I cannot see it. It must be improved.

Also regarding the presentation of the results in a table - form - it is still difficult to read. Please change the tabs 1 and 4 into the graph form - scientific graph with mean, max and min, deviation. In my opinion, it must be improved. Add SD and other statistics to the data given in tab4. To advance your statistics - you should use statistic testing methods. Then the "differences" will be statistically justified.

Response: thank you very much for your good observations and evaluation, which helps us to improve the manuscript in our future work, and we are very sorry for our incorrect or missing some essential of various parts of the manuscript. Answer. According to your comment we revised the Figs scale bar in figs 5 and 8. Please see details in lines (147-149 and 212-214). Furthermore, we changed the table 1 and table 4 into the graph form - scientific graph with mean and standared deviation and a nother statistics. Please see modefied tracking in lines (237,243,257,266, (274-280),282, 284, and lines (285-287)).

Note: Now Table 1 became Fig. 11, and Table 4 now became Fig. 12; besides that we added a scale bar to Fig. 5 and Fig. 8.

Reviewer 3 Report

Thank you for your responses. However, I am not satisfied before the SEM (and other) photos are not improved. Still, the scale bar (marker) should be added e.g. in the figs 5 and 8. Even the authors claim that it is given - I can not see it. It must be improved.

Also regarding the presentation of the results in a table - form - it is still difficult to read. Please change the tabs 1 and 4 into the graph form - scientific graph with mean, max and min, deviation. In my opinion, it must be improved. Add SD and other statistics to the data given in tab4. To advance your statistics - you should use statistic testing methods. Then the "differences" will be statistically justified.

Author Response

(The authors gave the same response as above.)
